# A Deep Learning Model for Detecting the Arrival Time of Weak Underwater Signals in Fluvial Acoustic Tomography Systems

**DOI:** 10.3390/s25030922

**Published:** 2025-02-03

**Authors:** Weicong Zheng, Xiaojian Yu, Xuming Peng, Chen Yang, Shu Wang, Hanyin Chen, Zhenxuan Bu, Yu Zhang, Yili Zhang, Lingli Lin

**Affiliations:** 1School of Information, Xiamen University, Xiamen 361102, China; zhengweicong@stu.xmu.edu.cn (W.Z.); wangshu@stu.xmu.edu.cn (S.W.); 2College of Biological Science and Technology, Minnan Normal University, Zhangzhou 363000, China; zhangyili20010214@163.com (Y.Z.); 13123174906@163.com (L.L.); 3Key Laboratory of Underwater Acoustic Communication and Marine Information Technology of the Ministry of Education, College of Ocean and Earth Sciences, Xiamen University, Xiamen 361005, China; pengxm@stu.xmu.edu.cn (X.P.); 22320240156729@stu.xmu.edu.cn (C.Y.); 25320231152507@stu.xmu.edu.cn (H.C.); 22320241151532@stu.xmu.edu.cn (Z.B.); yuzhang@xmu.edu.cn (Y.Z.); 4State Key Laboratory of Marine Environmental Science, College of Ocean and Earth Sciences, Xiamen University, Xiamen 361005, China; 5College of Ocean and Earth Sciences, Xiamen University, Xiamen 361005, China; 6School of Architecture and Civil Engineering, Xiamen University, Xiamen 361005, China

**Keywords:** fluvial acoustic tomography, feature extraction, attention mechanism, arrival time estimation

## Abstract

The fluvial acoustic tomography (FAT) system relies on the arrival time of the system signal to calculate the parameters of the region. The traditional method uses the matching filter method to calculate the peak position of the received acoustic signal after cross-correlation calculation within a certain time as the signal arrival time point, but this method is difficult to be effectively applied to the complex underwater environment, especially in the case of extremely low SNR. To solve this problem, a two-channel deep learning model (DCA-Net) is proposed to detect the arrival time of acoustic chromatographic signals. Firstly, an interactive module is designed to transmit the auxiliary information from the cross-correlation subnetwork to the original signal subnet to improve the feature information extraction capability of the network. In addition, an attention module is designed to enable the network to selectively focus on the important features of the received acoustic signals. Under the background of white Gaussian noise and real river environment noise, we use the received signals of the acoustic tomography system collected in the field to synthesize low SNR data of −10, −15, and −20 different decibels as datasets. The experimental results show that the proposed network model is superior to the traditional matching filtering method and some other deep neural networks in three low SNR datasets.

## 1. Introduction

Flow estimation is crucial in hydrological studies, extreme event analysis, and water resource management. Software modeling tools for flow and transport phenomena require detailed and accurate data. Pappenberger et al. [1] studied the impact of boundary condition uncertainty on flood inundation predictions and noted that uncertainty in inflow significantly affects model outputs. Therefore, a method for accurately measuring river flow is critical.

In the past, various acoustic instruments have been developed for continuous water flow measurements. Instruments like CTDs, portable depth and water level meters, and portable water temperature meters can only perform single-point measurements. Le Coz et al. [2] and Hoitink et al. [3] used Horizontal Acoustic Doppler Current Profilers (H-ADCPs) to collect velocity vectors on horizontal planes for continuous flow measurement. However, these devices cannot simultaneously capture velocities across entire cross-sections. For shorter sections, this might be acceptable, but for longer sections, the time intervals between data points can be very long, affecting real time data utility. Additionally, mobile observation methods are extremely dangerous and sometimes infeasible during severe weather conditions such as storms or typhoons. Therefore, accurately estimating flow in estuarine currents using these methods remains challenging.

Acoustic tomography is a technology that uses sound wave propagation characteristics for imaging and analysis and is widely used in medical imaging, underground structure exploration, and construction and civil engineering, such as using it to evaluate and identify the operating regime of strain hardening cementitious composites [4,5] and fast tomography, which can be used in the accelerated damage test of a steel fiber reinforced concrete beam [6]. Consequently, acoustic tomography technology [7,8,9,10,11] has garnered considerable interest among scholars due to its nonintrusive nature and ability to rapidly and conveniently acquire temperature and flow parameters over large observation areas. Using two-point acoustic tomography for long-term observation of hydrological parameters in shallow seas, estuaries, or channels involves the mutual exchange of signals between acoustic transducers at two stations, allowing the precise calculation of travel time and sound speed of signals between the two points. This enables the accurate reconstruction of average temperature and flow information in the monitored section, which, when combined with dynamic water levels and topography, allows for a precise calculation of river flow in the observed area. Consequently, sediment transport, pollutant dispersion, and total water quality parameters across a wide region can be estimated. Although the phase of acoustic tomography transmitted signals is modulated by a maximum length sequence (m-sequence) to improve the signal-to-noise ratio (SNR), the presence of natural conditions like river bottom characteristics, boundary reflections, and significant underwater noise can cause the SNR to drop sharply, hindering accurate arrival time calculations of acoustic signals at the receiving end and the affecting final speed and flow calculations. To improve the accuracy of detecting the arrival time of acoustic signals, some methods have been used for preprocessing the received signals. For instance, received acoustic signals are often affected by sudden underwater noise, resulting in spike data, and standard deviation, a simple method for removing spikes, has been used in fluvial acoustic tomography system studies. However, this cannot address the effects of afternoon phenomena. With the development of neural networks, deep learning models offer new approaches to solving these problems, providing strong adaptability and good generalization as a data-driven method.

## 2. Related Work

In the fields of underwater acoustic communication and acoustic tomography, detecting the arrival time of underwater signals has always been a key challenge. Traditionally, signal processing techniques such as matched filtering and receiver functions have been widely used for signal detection, but they perform poorly in noisy environments. Researchers have gradually attempted to employ more complex models to improve detection accuracy. Avik Kumar Das and colleagues used ICD to detect overlapping acoustic emission waves in real time, improving the detectability, reliability, and accuracy of conventional systems [12].

In recent years, deep learning models have shown great potential in handling complex signal detection tasks. Particularly in the areas of computer vision and speech recognition, deep learning technologies like Convolutional Neural Networks (CNN) and Long Short-Term Memory networks (LSTM) have achieved remarkable success. Consequently, some scholars have attempted to apply these technologies to signal detection tasks. Federica et al. [13] proposed using CNNs and a novel Capsule Neural Network to robustly detect the arrival time of sound signals in the air, replacing standard statistical strategies. Van-Thuan Tran et al. [14] introduced a new network model, MSNET, based on CNN, which outperformed all other baseline models and achieved the best detection results in train arrival detection events.

Aolong Zhou et al. [15] proposed a noise-aware deep learning model based on the full-band sub-band attention network (NAFSA-Net), which can denoise acoustic signals at low signal-to-noise ratios, significantly outperforming traditional methods and competing solutions based on DNN.

To the best of our knowledge, the direct Time of Arrival (ToA) estimation from long sequence input reception signals using supervised deep learning networks has not yet been studied in the literature. Based on the above research background, this study proposes a new deep learning model (DCA-Net) focused on detecting the arrival time of weak underwater signals in complex environments. Compared to existing methods, our approach aims to enhance detection accuracy and robustness under low signal-to-noise ratio conditions by directly obtaining the arrival time point of acoustic signals through the input of received signal sequences into a deep learning network, thereby better meeting the needs of practical applications.

## 3. Model and Experimental Design

Figure 1 presents the overall network architecture of the proposed model, which is a dual-channel deep learning detection model with an attention mechanism (DCA-Net). This network structure is mainly composed of residual blocks, an interaction module, and an attention module. The traditional method uses a threshold approach to perform cross-correlation between the original received acoustic signal sequence and the transmitted signal, taking the highest peak after calculating the absolute value as the time point of signal arrival. This approach neglects potential features hidden within the original signal sequence. To fully extract features from the original signal, we designed a dual-subnet network architecture. Our model takes the original received acoustic signal sequence as input, denoted as X∈R (T*1), and subsequently inputs it into the original signal subnet and the cross-correlation signal subnet. In both subnets, we stack multiple residual blocks to fully extract signal features and introduce an interaction module to transfer supplementary features from the cross-correlation subnet to the original signal subnet. In this way, it is expected that the original signal subnet will obtain complementary information. After passing through the attention module and the fully connected layer, the final detection result of the acoustic signal onset is outputted.

### 3.1. Subnetwork Module

In the proposed network architecture, the two subnetworks adopt the same residual block-stacking structure. The only difference is that the cross-correlation subnetwork adds a cross-correlation block before the residual blocks to receive the original transmitted signal, perform cross-correlation with the original received acoustic signal sequence, and convert it into a cross-correlation sequence. Subsequently, the same five improved residual blocks as in the original signal subnetwork are introduced to extract latent features. Each residual block consists of three convolutional layers, where the first convolutional layer has a kernel size of 1, and the other two have a kernel size of 15, all with a stride of 1. Unlike traditional residual blocks, the number of channels in the convolutional layers increases concerning their input. The purpose of deploying the first convolutional layer is primarily to increase the number of channels to ensure channel consistency in the skip connections. After the other two convolutional layers, there are batch normalization and activation unit blocks (ReLU and linear activation). We use X∈R (T*C) to denote the output of each residual block, where C is the number of channels. The output channel numbers for each residual block are 8, 16, 32, 64, and 128, respectively. After the skip connections, a max pooling layer with a pooling window size of 2 is deployed to perform pooling operations. Figure 2 shows the schematic diagram of the residual block in the subnet module.

### 3.2. Interaction Module

In our proposed model, the cross-correlation subnetwork is not independent of the original signal subnetwork. We aim for the cross-correlation subnetwork to enhance the training process of the original signal subnetwork. By performing cross-correlation calculations on the received acoustic signal sequence, the cross-correlation subnetwork can, to some extent, reduce the impact of noise. To this end, an information interaction module is introduced to transfer valuable information from the cross-correlation subnetwork to the original signal subnetwork. The structure of the interaction module is shown in Figure 3.

First, the features from the cross-correlation subnetwork are passed through a pooling layer. Then, we input the new features into a convolutional block composed of a one-dimensional convolutional layer and a sigmoid activation function to produce a mask M, which is used to filter the information passed into the original signal subnetwork. Finally, the features of the two subnetworks are concatenated to obtain the output features. The process of the interaction module is formulated as follows:$$M=MASK(F_{CRO}),F_{OUT}=F_{ORI}+F_{CRO}⊗M$$
where ⊗ denotes element-wise multiplication, and MASK (·) is shorthand for the pooling, convolution, and sigmoid functions.

### 3.3. Attention Module

Attention mechanisms were initially introduced to address sequence-to-sequence tasks, such as machine translation [16] and speech recognition [17]. Subsequently, they were extended to fields like computer vision [18,19,20,21] and audio signal processing [22,23,24], achieving great success.

Pandey and Wang [22] proposed a densely connected convolutional network with self-attention for speech enhancement, where the attention module was designed to capture dependencies at the utterance level. Additionally, researchers have suggested using channel-wise attention along the frequency dimension of the complex-valued spectrum to capture the structural characteristics of signals [25,26].

In the aforementioned approaches, attention mechanisms are typically applied to capture global context aggregation along the time dimension or frequency dimension. Unlike previous solutions, we propose using channel attention and spatial attention mechanisms to independently capture signal features, better acquiring features masked by noise.

Our proposed model captures more important features in the output by aggregating channel attention and spatial attention. The specific process involves passing subnetwork output features through the channel attention block followed by the spatial attention block. The structure of the attention module is shown in Figure 4.

### 3.4. Dataset

Referring to existing high-frequency fluvial acoustic tomography systems, we found that these systems use a relatively high transmission and reception sampling rate, such as 100 kHz. Therefore, even if the approximate time range of the acoustic signal arrival can be calculated through transmission and reception parameters and the distance between two points, the number of sequence points in the captured segment of the received signal can still be as long as 200 k. Additionally, the number of transmission signal points is typically above 5 k, so even if segmentations are made based on the received signal, the capture window generally needs to be over 20 k.

In our experiments, we use two types of background noise (Gaussian white noise and field-collected river environmental noise) and two types of target acoustic signals (actual received signals and simulated 7th-order m-sequence received signals) to generate a total of three experimental datasets (1. Gaussian white noise + actual received signal, 2. actual noise + simulated signal, 3. actual noise + actual received signal). The real data collection site is a section of the Jiulong River in Fujian Province, China. According to our topographic survey results, the maximum water depth within the observation area does not exceed 20 m, and the terrain generally features a deeper center with shallower sides. The vegetation on both banks of the river is relatively sparse, and the data collection period was concentrated in the autumn. At three typical signal-to-noise ratios (−10, −15, and −20 dB), three datasets were synthesized, each containing 4000 time sequence data with a length of 20,000 points, with 80% used for training, 10% for validation, and 10% for evaluation. On each dataset, we set different starting points for the target acoustic signals, from point 100 to point 4000, setting a starting point every 100 points, with 100 data entries for the same starting point. Figure 5 shows an example of the synthesized received signals of three datasets under −20 dB conditions.

### 3.5. Experimental Details

We used the Adam optimizer for optimization, with an initial learning rate of 5 × 10^−4^. The batch size was set to 16. Specifically, we employed learning rate scheduling, where the learning rate decreases by 90% if the evaluation on the validation set does not improve over five consecutive epochs, with a minimum learning rate set to 1 × 10^−6^. An early stopping mechanism was set with a patience of 10 epochs. The ratio for dividing our experimental training set, validation set, and test set was 8:1:1. All our code was implemented under the TensorFlow 2.4 framework, running on a computer equipped with an NVIDIA GTX 3070 graphics processor.

### 3.6. Performance Evaluation

To evaluate the specific performance of the network, we used the mean square error MSE and the mean absolute error MAE as objective indicators. MSE and MAE indexes are often used for objective evaluation of regression prediction in traditional algorithms and deep learning. Among them, MSE measures the average square variance between the predicted value and the actual value, which has an amplification effect on the deviation. This means that when the model has a large error, the MSE will give a higher penalty, forcing the model to fit the extreme values better. MAE provides an evaluation standard that treats all errors equally. The definition formulas for MAE and MSE are as follows.$$\mathrm{MSE}=\frac{1}{n}{\sum_{i=1}^{n}\left(y_{i}-{\hat{y}}_{i}\right)}^{2},\mathrm{MAE}=\frac{1}{n}\sum_{i=1}^{n}\left|y_{i}-{\hat{y}}_{i}\right|$$

Here, *n* represents the total number of samples, *i* represents the *i*-th sample, $y_{i}$ represents the label value of the *i*-th sample, and ${\hat{y}}_{i}$ represents the predicted value for the *i*-th sample.

## 4. Result

### 4.1. Ablation Experiments

In this section, we present the results of the ablation experiments to analyze the effectiveness of different network modules in our proposed DCA-Net. Five groups of comparative experiments were designed to evaluate the performance of different modules. First, we compared two baseline models: a single-channel network and a dual-channel network (both without deploying interaction modules and attention modules). Then, an interaction module was deployed between the dual-channel networks, and the attention module was added. Finally, both the interaction module and attention module were deployed to obtain our proposed DCA-Net. All models use the MSE as the loss function to optimize the training process and the MAE function as the evaluation metric. The results of the ablation experiment on the −15 dB real transmission signal + Gaussian white noise dataset are shown in Figure 6.

In the above figure, TM represents the traditional matched filtering method, SCB represents a simple model with only a single channel, DCB represents a dual-channel model, DCB+INTERACTION represents the dual-channel model with an interaction module, DCB+AM represents the dual-channel model with an attention module, and DCA-Net represents the final model we proposed. From the above figure, it can be seen that compared with the traditional matched filtering method, the simple single-channel network model alone can reduce the mean absolute error (MAE) of the detection results by 33.75, while the dual-channel baseline model reduces the detection MAE by 85.62. After deploying the interaction module between the original signal subnet and the cross-correlation subnet, we observed an additional MAE reduction of 39.2. When introducing the attention module into the network without deploying the interaction module, the observed MAE reduction was 65.77. Finally, deploying both the interaction module and attention module in the network yields our proposed DCA-Net, which enhances the MAE reduction by 79.53 and 52.96, respectively, compared to adding the interaction and attention modules separately. These observations validate the effectiveness of the proposed dual-channel network architecture, interaction module, and attention module in detecting the arrival time of acoustic tomography signals under low signal-to-noise ratio conditions. Additionally, we can see that the dual-channel model with the attention module exhibits very competitive detection performance gains.

Figure 7 shows the MSE evaluation metric for the ablation experiments under a −15 dB signal-to-noise ratio. It can be seen that even our baseline model, compared to traditional detection methods, not only reduces prediction bias but also demonstrates greater stability in predictions according to the MSE results.

### 4.2. Model Comparison

In this section, we will analyze the results of different deep learning models to demonstrate that DCA-Net outperforms other deep learning models and traditional detection techniques in detecting the arrival time of acoustic tomography signals. Traditional detection techniques directly detect peak points as signal arrival time points after performing cross-correlation calculations on received signals. For end-to-end deep learning models, we can directly input the segmented received signals into the model to output the predicted signal arrival time points. Considering the length of data sequences, some complex models demand significant computational resources, so we selected some representative classical deep learning models for comparison with DCA-Net.

Table 1 shows the results of different models on the −10 dB, −15 dB, and −20 dB Gaussian white noise + actual transmission signal datasets. It can be seen that various models modified by different deep learning methods outperform traditional detection algorithms in terms of mean absolute error under three typical signal-to-noise ratios. Additionally, our proposed DCA-Net (last row) achieved the best evaluation scores on this dataset, significantly outperforming other models. DCA-Net achieved mean absolute error reductions of 201.73, 204.35, and 232.62 under −10 dB, −15 dB, and −20 dB signal-to-noise ratios, respectively.

Figure 8 shows the detection results of various deep learning models, including our proposed DCA-Net, for the same sample under a −20 dB signal-to-noise ratio. The purple dashed line represents the signal onset detected by the traditional algorithm, the red dashed line corresponds to the onset detected by our proposed DCA-Net, and the green dashed line is the true label value. The other colored dashed lines represent the prediction results of different models based on deep neural networks. It is evident that in degraded signals with low signal-to-noise ratios, the target acoustic signal is completely masked by environmental noise. This is especially true for high-energy impulsive noise in river environments, which renders traditional detection methods almost entirely ineffective at detecting the target acoustic signal. Although previous deep neural network-based techniques, such as CNN, WaveNet, and ResNet, perform better than traditional detection techniques, their detection effectiveness is still suboptimal under low signal-to-noise ratio conditions. In contrast, our proposed DCA-Net employs a dual-channel network architecture to capture more signal features and, combined with interaction modules and attention modules, extracts effective features for precise detection.

To further explore the detection performance of our proposed DCA-Net, we additionally compared the detection results of different DNN-based techniques on simulated signal + real noise synthetic datasets and real signal noise synthetic datasets. Table 2 shows the detection results of different techniques on a 7th-order m-sequence simulated signal + real noise synthetic dataset. It can be observed that compared to using real received signals, the 7th-order m-sequence provides a gain in detecting the acoustic signal onset. Nonetheless, our proposed DCA-Net still achieves the best detection results.

Table 3 presents the detection results of traditional detection techniques and various neural network-based techniques on real datasets sampled in the field. According to the distribution of target acoustic signal onset set in our dataset, it can be seen that when the signal-to-noise ratio is below −10 dB, the performance of traditional detection techniques deteriorates sharply. Classical deep learning methods such as CNN, on the other hand, exhibit smaller deviations in detection results compared to traditional techniques. Our proposed DCA-Net achieved the best results, reducing the mean absolute error by 920.46, 1716.95, and 1801.66 at −10 dB, −15 dB, and −20 dB, respectively, compared to traditional techniques. Compared with other classical deep learning-based techniques, our proposed model also achieves varying degrees of improvement in detection error.

## 5. Conclusions

In this paper, DCA-Net, an end-to-end deep learning model for detecting underwater acoustic signal arrival time under low SNR conditions, is introduced. The scheme uses a two-channel network structure to improve the feature extraction of the received sound signal, and the cross-correlation subnetwork is designed to assist the training of the original signal subnet. DCA-Net was trained and tested by constructing three datasets, which were composed of Gaussian white noise, real stream acquisition noise, simulated 7th-order M-sequence emission signal, and real emission acquisition signal. Through the model ablation experiment on the real signal + Gaussian white noise dataset, we found that compared with the single-channel network structure with stacked residuals, the dual-channel structure promoted the model prediction results to achieve more than 10% gain on both MSE and MAE indexes, which has a greater advantage for the feature extraction of one-dimensional underwater acoustic signals. In addition, in this dataset, we also evaluate the detection effect gain of the interaction module and the attention module in the whole network. Based on the two-channel architecture, the deployment of the interaction module provides a 13% average absolute error detection gain for the model, while the attention module provides a 22% detection gain.

We evaluated DCA-Net under three datasets with typically low SNR. It can be seen that under real environment conditions, when the SNR is lower than −10 dB, the traditional matching filtering method can hardly detect the starting point of the target acoustic signal. The simulated transmitted signal and Gaussian white noise have better detection accuracy than the real signal and noise under the same signal-to-noise ratio. Our proposed DCA-Net has better detection accuracy in all datasets than traditional matching filtering methods and other deep learning models.

## Figures and Tables

**Figure 1 sensors-25-00922-f001:**
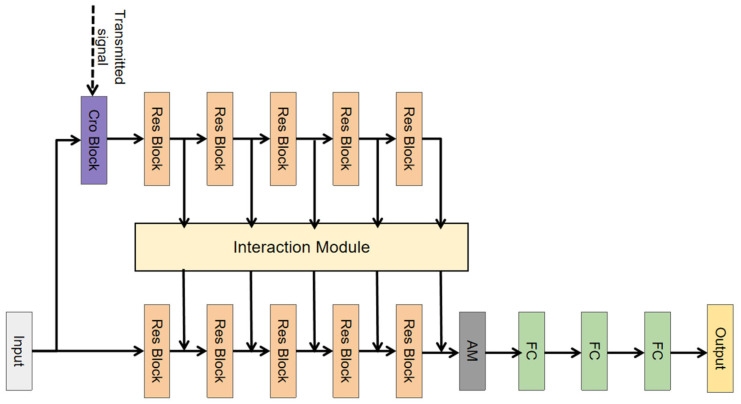
Network architecture diagram.

**Figure 2 sensors-25-00922-f002:**
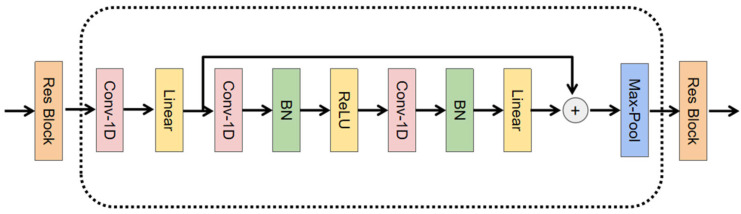
Res Block Structure Diagram.

**Figure 3 sensors-25-00922-f003:**
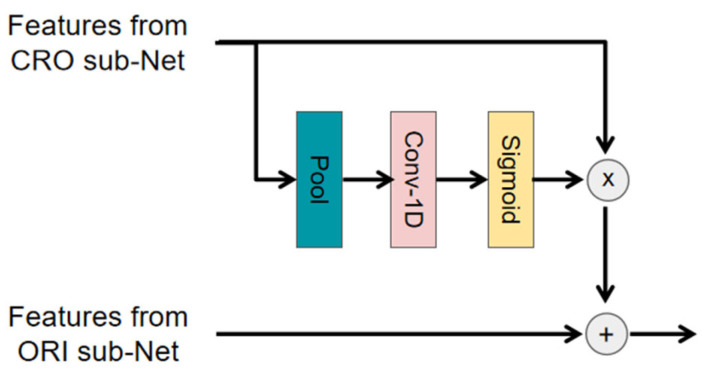
Interaction Module Structure Diagram.

**Figure 4 sensors-25-00922-f004:**
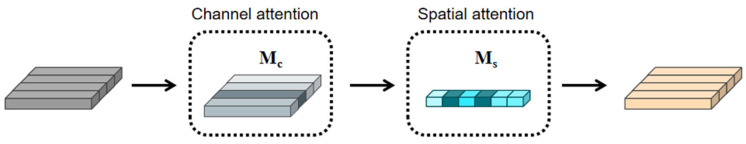
Attention Module Structure Diagram.

**Figure 5 sensors-25-00922-f005:**
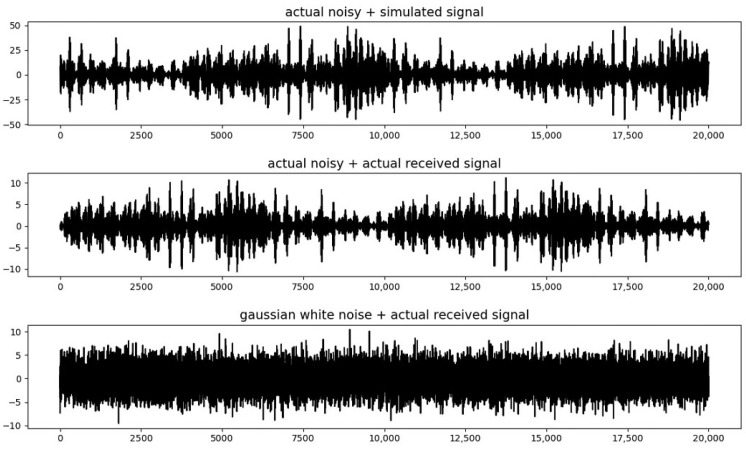
The synthesized received signals.

**Figure 6 sensors-25-00922-f006:**
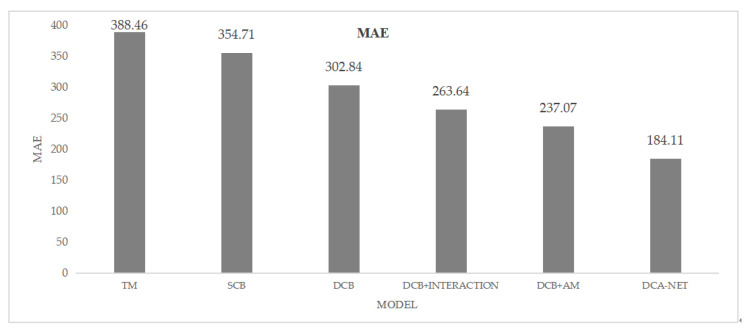
Ablation Result (MAE).

**Figure 7 sensors-25-00922-f007:**
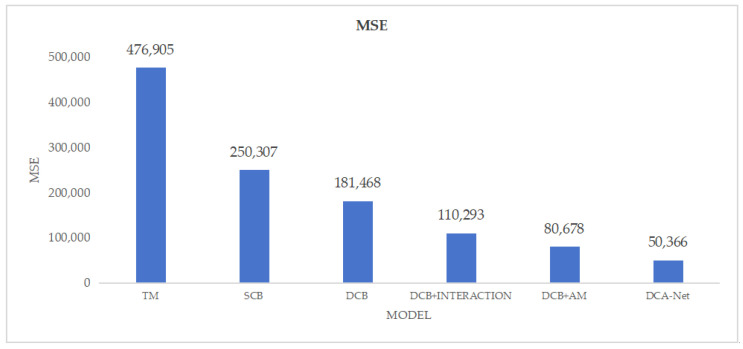
Ablation Result (MSE).

**Figure 8 sensors-25-00922-f008:**
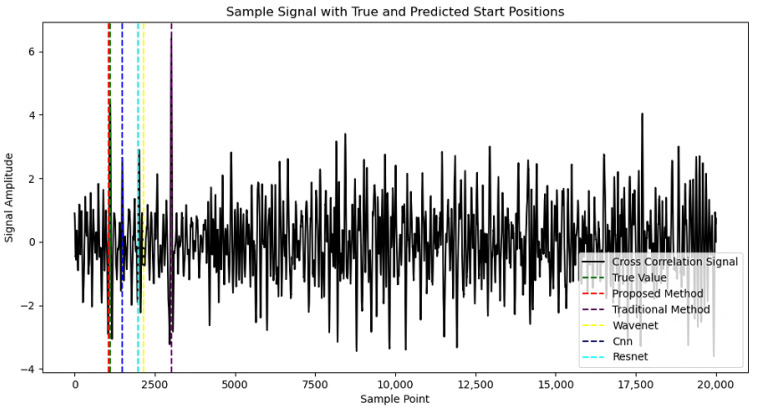
Comparison of Results for Each Model.

**Table 1 sensors-25-00922-t001:** Experimental results for Gaussian white noise + actual transmission signal datasets.

	−10 dB	−15 dB	−20 dB
Traditional Method	354.72	388.46	568.97
Wavenet	240.35	276.42	392.72
Cnn	269.06	337.52	442.13
Resnet	328.33	341.99	442.81
DCA-Net	**152.99**	**184.11**	**336.35**

**Table 2 sensors-25-00922-t002:** Experimental results for simulated signal + real noise synthetic datasets.

	−10 dB	−15 dB	−20 dB
Traditional Method	170.11	195.07	291.37
Wavenet	213.18	237.08	299.17
Cnn	195.69	223.42	278.40
Resnet	250.85	314.76	350.55
DCA-Net	**138.93**	**184.11**	**236.84**

**Table 3 sensors-25-00922-t003:** Experimental results for real datasets.

	−10 dB	−15 dB	−20 dB
Traditional Method	1125.20	2030.82	2195.52
Wavenet	410.09	641.47	1011.33
Cnn	384.01	604.96	973.15
Resnet	425.60	470.81	649.02
DCA-Net	**204.74**	**313.87**	**393.86**

## Data Availability

All data supporting this study are available upon request.

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
