# Peer review of "A Deep Learning Model for Detecting the Arrival Time of Weak Underwater Signals in Fluvial Acoustic Tomography Systems"

_sensors, 2025, doi:10.3390/s25030922_

Round 1
Reviewer 1 Report
Comments and Suggestions for Authors
My comments are attached within the manuscript.

Please recheck the grammar and sentences once more.
Reviewer 2 Report
Comments and Suggestions for Authors
Comments are presented in the attached file "review-comments.docx".

Round 2
Reviewer 1 Report
Comments and Suggestions for Authors
Thank you for improving the manuscript. I am afraid I sent the wrong DOI. The correct DOI is as follows. Please correct the previous unresolved comment.
https://doi.org/10.1177/14759217211036881
Author Response
Comments 1: [Thank you for improving the manuscript. I am afraid I sent the wrong DOI. The correct DOI is as follows. Please correct the previous unresolved comment.
https://doi.org/10.1177/14759217211036881]
Response 1: [Thank you for your suggestions. We have revised the previous unresolved comments based on your suggestions. This change can be found on page 2 line 57,58.]
Reviewer 2 Report
Comments and Suggestions for Authors
This is an improved version of the manuscript. I can see the authors have made efforts to address my concerns in the revised manuscripts. They are acceptable.
Author Response
Comments 1: [This is an improved version of the manuscript. I can see the authors have made efforts to address my concerns in the revised manuscripts. They are acceptable.]
Response 1: [Thank you for your recognition. Your acknowledgment greatly motivates us, and we will continue to work hard.]